# Human Breast Milk miRNAs: Investigation of Association Between Breastfeeding Children and Maternal Obesity in Obesity Development in Offspring

**DOI:** 10.3390/genes16111373

**Published:** 2025-11-11

**Authors:** Marina Chondrogianni, Maria Lithoxopoulou, Athina Ververi, Alexandros Lampropoulos, Alexandros Sotiriadis, Eystratios Kolibianakis

**Affiliations:** 1School of Medicine, Aristotle University of Thessaloniki, 54124 Thessaloniki, Greece; marinachon20@gmail.com (M.C.); lithoxopoulou@auth.gr (M.L.); lambrop@auth.gr (A.L.); alesoti@auth.gr (A.S.); kstratis@auth.gr (E.K.); 2Second Department of Neonatology & NICU, School of Medicine, Aristotle University of Thessaloniki, 54124 Thessaloniki, Greece; 3Department of Genetics for Rare Diseases, Papageorgiou General Hospital, 56429 Pavlos Melas, Greece; 4Department of genetics, Papageorgiou General Hospital, 56429 Pavlos Melas, Greece; 5First Department of Obstetrics and Gynecology, Faculty of Medicine, School of Health Sciences, Aristotle University of Thessaloniki, 54622 Thessaloniki, Greece; 6Unit for Human Reproduction at the 1st Department of Obstetrics and Gynecology, School of Medicine, Aristotle University of Thessaloniki, 54124 Thessaloniki, Greece

**Keywords:** breast milk, lactation, breastfeeding, miRNAs, epigenetics, obesity, maternal obesity

## Abstract

Background/Objectives: Human breast milk is a mammary gland secretion with a dynamic composition, containing important bioactive factors for infant growth. Epigenetic factors, like microRNAs, are found in breast milk and can regulate gene expression and, thus, infant growth. Obesity is, among others, a major global health concern with long-term consequences, making its prevention during early life a public health priority. Maternal lifestyle factors, including diet and body weight status, may influence infant growth patterns and susceptibility to obesity. The aim of this review is to explore the hypothesis that miRNA content in breast milk might be influenced by maternal obesity, eventually affecting the obesity risk in offspring. Methods: This systematic review was carried out in line with the PRISMA 2020 statement and included observational (cohort) studies that met the inclusion criteria and compare the expression of miRNAs in OW/OB lactating mothers and associate this to the obesity development in the offspring. Results: According to the included studies, the most common miRNAs are miR-148a, miR-30 family, and miR-let7 family, with miR-30b and miR-let7a among the most discussed that participate in adipogenesis. Some of these miRNAs secreted in breast milk pass on a genetic predisposition for obesity to the next generation, while others provide a protective role against obesity in the offspring. Conclusions: Eventually, even though individual miRNAs may fluctuate, the overall miRNA profile remains stable. The findings underscore the importance of balanced maternal nutrition and optimal health during lactation, both for supporting healthy infant development and for potentially reducing the risk of obesity later in life.

## 1. Introduction

### 1.1. Human Breast Milk Benefits

Human breast milk is globally recognized for its benefits and as the most suitable source of infant nutrition. Breast milk is not a homogeneous body fluid, but rather a mammary gland secretion whose consistency and dynamic composition vary during the day and each lactation meal. Mature human breast milk includes a diverse range of constituents from fat and proteins to carbohydrates and minerals. Moreover, it is already known that breastfeeding enhances the immune system of the infant and resistance to infectious diseases [1,2,3]. This is due to the various components found in breast milk, such as antibodies and, specifically, Immunoglobulin A (IgA). These antibodies participate in the prevention of illnesses by neutralizing pathogens in mucous membranes of the infant’s throat, lungs, and intestines. Apart from that, white blood cells are included in human breast milk, too, providing a direct defense against infectious agents [4]. However, the most remarkable feature of human breast milk is its composition of bioactive components. These components provide antimicrobial protection and help in improving the health and maturation of immune cells, building a stronger immune system [5,6].

Studies have shown consistently that breastfed infants have fewer infections of the respiratory and gastrointestinal systems. The protection afforded by breastfeeding is proven to have benefits not only during infancy but also throughout the whole life, and decreased development of certain health conditions, such as Type 1 diabetes (T1D), celiac disease (CD), and some allergies [7].

### 1.2. Childhood Obesity Risks

Obesity is a major health condition among adults and children. Nowadays, it is believed that obesity is of multifactorial origin, influenced by various social and environmental factors [8]. The main issue with obesity or overweight in early years is that childhood and infancy obesity is very likely to remain a concern until adulthood. This persistent weight excess predisposes infants to the early development of a variety of health complications in later life. One of them is high blood pressure, leading to hypertension, which in turn increases the risk for more complicated cardiovascular problems, like coronary heart disease (CHD), one of the most common causes of death worldwide [9,10]. Additionally, the presence of excess visceral fat along with high blood pressure elevates the chances of developing metabolic syndrome (MetS). This, consequently, increases the risk for cardiovascular diseases (CVD) and Type 2 diabetes (T2D). According to the above, interventions and early prevention make it essential to address childhood obesity [11].

### 1.3. Epigenetic Regulators in Human Breast Milk

Human breast milk, as a body fluid rich in bioactive components, also contains epigenetic regulators, which may control gene expression and eventually have an impact on the growth of children. Key epigenetic regulators include microRNAs (miRNAs), DNA methylation and histone modification regulators, long non-coding RNAs, exosomes, growth factors, and hormones [12].

MiRNAs are small non-coding RNAs that are essential to the post-transcriptional control of gene expression. Multiple miRNAs have been detected in breast milk and can be passed to the offspring. These miRNAs have been associated with immunological modulation, development, and even long-term metabolic impacts [13].

MiRNAs found in breast milk are usually encapsulated in exosomes (or extracellular vesicles, EVs, or breast milk extracellular vesicles, bEVs) along with other mRNAs, proteins, or small molecules. These exosomes protect them from degradation, promote their transport between cells, and can transmit epigenetic information from one generation to the next, altering gene expression and development [14]. Breast milk also contains growth factors (such as TGF-β) and hormones (for instance, insulin and leptin) that influence gene expression by binding to specific binding sites in the offspring. These substances may stimulate signaling pathways that result in epigenetic changes like DNA methylation and histone modification [6].

### 1.4. Vertical Transmission of Milk Exosomal MicroRNAs

There are two opposing theories that explain the biological significance of milk miRNAs. The nutritional hypothesis considers miRNAs as degradable nutrients that can provide nucleotides without specific signaling activity [15]. In contrast, the functional hypothesis proposes that miRNAs, particularly those enclosed in small extracellular vesicles (sEVs), are capable of regulating gene expression in the infant [16].

Exosomal encapsulation of milk miRNAs provides exceptional protection and resistance to digestion and enzymatic degradation through the gastrointestinal tract [15,17]. This allows sEVs to be absorbed by intestinal epithelial cells, as demonstrated using neonatal enteroid models [18]. Complementary in vivo tracer studies have confirmed that milk-derived miRNAs traverse the intestinal barrier and are incorporated into Argonaute-2 (AGO2) complexes, a hallmark of functional engagement in post-transcriptional gene regulation [19].

Beyond direct host regulation, milk sEVs also influence the gut microbiota. They enhance the growth of Bifidobacterium through metabolic modulation [20], and contribute to intestinal microbial balance with potential benefits for infants [17]. Collectively, these findings support the idea that milk is not simply a nutritional fluid but an advanced signaling system, transmitting maternal regulatory signals that may shape early development and immunometabolic programming [15,16,17,18,19,20].

### 1.5. Epigenetic Aspects of Adipogenesis

#### 1.5.1. The Wnt Signaling Pathway

Human adipogenic differentiation is controlled by the Wnt/β-catenin signaling pathway, which is responsible for the conversion of mesenchymal cells into mature fat cells (adipocytes). When the signaling pathway is active, β-catenin antagonizes peroxisome proliferator-activated receptor γ (PPARγ) and C/EBP family members, suppressing the adipogenic transcriptional program. Conversely, attenuation of Wnt/β-catenin signaling allows these adipogenic regulators to be expressed and leads to adipocyte commitment and lipid accumulation [21,22,23,24]. This central role of Wnt as a basic regulator makes the pathway a natural focus for considering the epigenetic influence of progenitor cells on adopting adipogenic vs. osteogenic or other mesenchymal forms.

Epigenetic modifications, such as DNA methylation, histone modifications, and RNA-mediated post-transcriptional control, are influenced by Wnt pathway activity during adipogenesis. DNA methyltransferases (DNMTs) catalyze methylation of cytosine at gene promoters and regulatory elements, which can repress gene expression, including components or regulators of Wnt signaling. Similarly, RNA modifications and RNA-binding proteins—for instance, m^6^A dynamics mediated by FTO (fat mass and obesity-associated protein)—modulate mRNA stability and translation of adipogenic regulators, thereby shaping lipid metabolism and differentiation outcomes [25]. MicroRNAs represent another layer of epigenetic control: by base-pairing with target mRNAs, they reduce translation or promote degradation of transcripts encoding signaling modulators, chromatin regulators, or transcription factors relevant to adipogenesis [26,27].

Within this framework, milk-derived exosomal miRNAs—most notably, the miR-148a and miR-22 species identified in human milk studies—are positioned as potential modulators of Wnt-dependent lipogenic programming [26,28]. Functionally relevant connections between these miRNAs and Wnt epigenetic regulators have been reported. MiR-148a is a well-established inhibitor of DNMT1 expression and has been experimentally shown in some contexts to affect the activity of the Wnt pathway [16,26], while experimental manipulation of miR-148a-3p in other cell types can relieve the inhibition of Wnt signaling by targeting extracellular Wnt antagonists (for example, DKK1), thereby promoting Wnt-driven differentiation programs [29]. Members of the miR-22 family have also been involved in pathways intersecting with chromatin regulators and signaling hubs related to cell fate decisions [16,26,27].

Taken together, these observations allow a feasible model based on the following: maternal milk exosomal miRNAs administered to the infant could modify the epigenetic landscape of mesenchymal progenitors by repressing DNMTs, and thereby reducing promoter methylation at Wnt pathway genes or regulatory elements, and targeting mRNAs encoding specific Wnt inhibitors, leading to an altered tone of Wnt signaling.

Either mechanism would shift the balance of Wnt activity and, thus, alter the threshold for lipogenic engagement. This model is consistent with evidence indicating that perinatal environmental factors and nutritional exposures modulate the lipogenic capacity of progenitor cells. For example, mesenchymal stem cells derived from neonates of obese mothers exhibit an enhanced propensity for lipogenesis [30]. Moreover, it aligns with the broader hypothesis that compositional differences between breast milk and infant formula - particularly the absence of specific exosomal microRNAs in formula - may underlie divergent epigenetic regulation of Wnt signaling pathways, ultimately influencing lipogenic programming and metabolic outcomes [28,31].

It is important to emphasize the current limits of the evidence: while exogenous miRNAs from human milk are stable and bioavailable, and while individual miRNAs such as miR-148a can target DNMTs or Wnt antagonists in experimental systems, direct proof that miR-148a or miR-22-3p in breast milk causally remodel Wnt promoter methylation in human infant mesenchymal progenitor cells, and thereby increase adipocyte production in vivo, remains unclear at present.

#### 1.5.2. The FTO Protein

The FTO gene encodes an N^6^-methyladenosine (m^6^A) RNA demethylase that has emerged as a critical post-transcriptional regulator of adipocyte differentiation and lipid metabolism. FTO catalyzes the removal of m^6^A modifications from mRNAs, thereby affecting transcript stability, splicing, and translational efficiency [25,32]. Increased FTO expression in mesenchymal and preadipocyte cells enhances adipogenic commitment through the upregulation of key transcriptional regulators such as PPARγ and C/EBPα. By demethylating m^6^A residues within these mRNAs, FTO increases their half-life and translation, facilitating the transition from progenitor to mature adipocyte [32].

In addition to its role in controlling lipogenic transcription, FTO also regulates lipid metabolism homeostasis. It promotes the maturation of sterol regulatory element binding protein 1c (SREBP1c), and induces transcription of the cell-death-inducing DFFA-like effector C (CIDEC), two mechanisms that enhance triglyceride synthesis and lipid droplet accumulation [33]. Collectively, these mechanisms position FTO as a central hub linking RNA methylation status to lipid storage and adipocyte function.

Recent data further suggest that FTO activity may interact with milk-derived epigenetic regulators in early life. In breastfed infants, exosomal miRNAs such as miR-148a-3p and miR-22-3p—which target DNMT1 and DNMT3A—help maintain a balanced DNA methylation environment that supports Wnt/β-catenin signaling, thereby opposing premature lipogenic differentiation [28,34]. In contrast, the absence or significantly reduced levels of these miRNAs in infant milk [35,36] may lead to DNMT-mediated hypermethylation of Wnt promoters, weakening anti-lipogenetic signaling while allowing FTO-driven m^6^A demethylation of adipogenic and lipogenic transcripts. This synergistic dysregulation could favor adipocyte expansion and lipid accumulation, contributing to the programming of obesity in formula-fed infants [34].

Thus, FTO represents a central epitranscriptomic mediator that integrates nutritional stimuli, DNA methylation dynamics, and m^6^A-dependent post-transcriptional regulation. Its function exemplifies how early-life nutritional exposures can exert long-term metabolic effects through coordinated epigenetic and epitranscriptional mechanisms.

### 1.6. Epigenetic Impact of Maternal Lifestyle and Nutrition During Pregnancy and Lactation in Children

Maternal lifestyle and nutrition may have a significant epigenetic impact on fetal development during pregnancy and breastfeeding. Nearly the same epigenetic mechanisms impact mother and fetus, as well as breast milk.

Maternal nutrition is the most important way in which epigenetic regulatory factors impact the fetus. Sufficient intake of folate, choline, vitamin B12, and other B vitamins is essential for adequate DNA methylation. These nutrients are vital for neural development and the regulation of metabolism. Furthermore, omega-3 and omega-6 fatty acids can influence gene expression via epigenetic mechanisms. As an example, maternal PUFA intake has been linked to alterations in patterns of DNA methylation in genes related to inflammation and metabolism in children [37,38,39].

Maternal consumption of polyphenol-rich foods (i.e., fruits, vegetables, and tea) may transfer bioactive compounds into maternal milk, which can modulate the epigenetic regulation of genes related to growth and metabolism. Lacking protein intake during pregnancy can result in alterations to histone modification and DNA methylation, possibly leading to delayed development and metabolic disorders in offspring [40,41].

Maternal stress impacts the growing fetus as well. Significant levels of stress during pregnancy can raise cortisol levels, which can cross the placenta and influence fetal development. High levels of maternal stress can also cause a rise in cortisol in breast milk, disrupting the infant’s stress-response system via epigenetic changes. This exposure can affect DNA methylation and histone changes in the embryonic brain, affecting genes linked to stress responses and elevating the possibility of neurodevelopmental issues [42]. Chronic stress can cause epigenetic programming of the hypothalamic–pituitary–adrenal (HPA) axis in the developing baby, increasing the child’s vulnerability to stress and anxiety disorders throughout their life [43].

Environmental pollutants, such as BPA and heavy metals, can alter typical epigenetic processes in the developing embryo and be passed to the newborn via breast milk. Such exposure may raise the risk of developmental defects and chronic health conditions [44]. Chemicals that disrupt hormone signaling can also have an impact on the inheritance of the epigenetic profile of genes involved in growth and development. In particular, being exposed to environmental pollutants during pregnancy has been associated with abnormal methylation patterns of DNA in children, which might result in reproductive and metabolic problems [45].

While many epigenetic regulators have an adverse effect on children, physical activity while pregnant has been linked to advantageous epigenetic alterations. Exercise may impact DNA methylation and histone acetylation in genes associated with metabolism and inflammation, possibly decreasing the chance of being overweight and obesity-related conditions in children [46].

In the end, sleeping habits and circadian rhythms can epigenetically affect the fetus’s DNA, altering clock gene expression and increasing sensitivity to metabolic and sleep problems [47]. Therefore, early-life nutrition not only provides calories but also actively programs children’s genes via epigenetics, giving a lifelong imprint on the health of the offspring.

While there is limited data on how maternal lifestyle directly affects miRNA expression in breast milk, emerging evidence suggests that maternal nutrition and environmental exposures can alter the composition of body fluids, including milk. Maternal obesity, in particular, is suspected to influence both the concentration and function of miRNAs, although more research is needed to fully understand these mechanisms.

Finally, maternal obesity is a factor that can probably influence miRNA content and expression levels. Although many studies have been performed previously about maternal obesity and its effect on the offspring in both animals and humans, little is known so far about the genetic inheritance of obesity and the miRNA profile that is transmitted to the next generation via breastfeeding. This systematic review investigates how human breast milk miRNAs are influenced by maternal obesity, eventually affecting obesity risk in the offspring.

## 2. Materials and Methods

This systematic review was carried out in line with the guidelines of the PRISMA (Preferred Reporting Items for Systematic Reviews and Meta-Analyses) 2020 statement. PubMed, Cochrane Library, Scopus, and Google Scholar were searched in July 2024. Additionally, a hand search was conducted in the references of the included studies.

**Table 1 genes-16-01373-t001:** Inclusion and exclusion criteria of this systematic review.

Inclusion Criteria	Exclusion Criteria
Randomized controlled trials, cohort studies, observational studies, case reports.	Reviews, meta-analyses, or any other type of study.
Studies performed in humans.	Studies not performed in humans.
Studies conducted on breastfeeding women.	Studies including men and women.
Free full texts.	Inaccessible articles.

This systematic review includes observational (cohort) studies that meet the inclusion standards, as represented in Table 1 and compare or make clear the difference in the expression of miRNAs in OW/OB lactating mothers and associate this with obesity development or the adipogenesis-related pathway in offspring.

A pre-organized spreadsheet was used for the data collection of the included studies. The extracted data were organized and sorted based on the following categories: (1) author and publication date; (2) type of study; (3) sample characteristics (population and sample analyzed); (4) method of miRNA analysis; (5) maternal characteristics: health status, BMI, and health issues (GDM, T1D, hypertension, preeclampsia, etc.); (6) infant characteristics: (i) breastfeeding status, (ii) gestational age and gender. The general characteristics of the included studies are presented in Table 2.

## 3. Results

### 3.1. General Results of the Study Process

The initial search retrieved a total of 33,831 results. Applying filters and title screening resulted in an overall number of 121 studies. The eligibility assessment involved the exclusion of duplicates, studies that did not meet the inclusion criteria, or irrelevant studies—such as studies not mentioning OW/OB lactating mothers, not analyzing lactating mothers separately based on their body mass index (BMI) to correlate alterations between miRNA expression, and studies failing to associate their results to maternal BMI or having significant bias. The entire selection procedure gave a total of seven studies, as presented in Figure 1.

One study was published in 2015, and the rest after 2021. The specimen analyzed in all studies was human breast milk (foremilk, hindmilk, colostrum, or mature milk), except one that investigated the urine and plasma of preterm infants as a result of consumption of human breast milk. Three studies undertook a comparison between OW/OB and NW mothers. Two studies examined mother–infant pairs lacking comparison. A single study assessed colostrum and mature milk, while a different one contrasted breast-fed infants with formula-fed. All of them studied the associations between maternal characteristics/BMI and the expression levels of miRNA in human breast milk, along with possibly in infant urine or plasma. Four studies included infant anthropometric measurements, with milk collection and infant measurements performed at different time points in each study. The miRNA method of analysis differed across the studies, with two of them using real-time PCR, one RT-qPCR, two small-RNA sequencing and RNA library preparation, one miRNA WTA, and one using the Nanostring nCounter method of miRNA analysis. In total, five studies investigated the associations between typical overlapping sets of miRNAs, whilst one study resulted in a different outcome compared to the rest.

### 3.2. General Characteristics of Included Studies

The first study involved in this review looked at the associations between breast milk miRNAs and maternal characteristics, performed by Xi Y. et al. in 2015 [50]. They compared colostrum (collected 2–5 days post-partum) and mature milk (3 months) obtained from a total of 119 nursing mothers (including mothers with hypertension and GDM). The aim of the study was to investigate the factors that influence the expression of specific breast milk miRNAs (miR-let-7a, miR-30b, and miR-378), already known to play a role in adipogenesis. Associations between miRNAs and maternal characteristics like breastfeeding duration, maternal BMI, and maternal/gestational age and infant gender were also investigated. Zamanillo R. et al. [48], a few years later, studied the associations between miRNAs in human breast milk, leptin, adiponectin, and infant weight. Breast milk was collected from 59 lactating mothers at 1, 2, and 3 months after birth, while anthropometric measurements followed infants until 2 years of age. The study’s purpose was to investigate the expression and associations of selected miRNAs known to play crucial roles in adipogenesis and obesity.

A cohort study performed in 2021 by Shah K.B. et al. [49] included a subset of 60 lactating women from a larger group of 365 mother–infant pairs. Milk was obtained at the first and third months of lactation, and during the last, milk was obtained from a subset of 48 initially enrolled women, while anthropometric measurements were performed at the first, third, and sixth months post-partum; this pilot study investigated the expression levels and associations of breast milk miRNAs and infant development between OW/OB and NW mothers during the first 6 months of age. Kupsco A. et al. [51], the same year, explored in a prospective study the association between different sets of miRNAs and maternal characteristics in the Faroe Islands; this study of human breast milk miRNAs, including 364 mother–infant dyads, had been the largest so far. Additionally, mothers who used to smoke during pregnancy, preeclamptic women, and mothers with GDM and T1D were also included. Next year, Eun Y. Cho et al. [52] used a cohort of 65 lactating women to compare bEV miRNAs of OW/OB to NW mothers.

Finally, the most recent studies were performed by Kim E.-B. et al. [53], who studied miRNA expression in infant urine and plasma after lactation during their stay in the NICU, and from Van Syoc E. et al. [54], who used 163 mother–infant pairs—excluding mothers with co-morbidities interfering with BF—and investigated the correlation between breast milk miRNAs and infant obesity. Foremilk was collected at birth, in the 1st and 4th months of lactation, while infant anthropometric measurements were collected at birth, and the 1st, 4th, 6th, and 12th months of age.

### 3.3. MiRNAs in Human Breast Milk Related to Maternal Obesity and Obesity Development in Offspring

#### 3.3.1. Role of miRNAs in Obesity Development

Table 3 shows the aforementioned miRNAs that are most abundantly expressed in EVs, mostly in breast milk, except one study that studied EVs in urine and serum derived from premature infants and their role in adipogenesis and obesity, taking into account maternal BMI. The majority of miRNAs were chosen due to their high expression levels in breast milk, or the fact that they are targets of leptin, adiponectin, or their receptors (miR-30a, miR-146b, miR-17, miR-let-7a, and miR-222), or are somehow related to adipogenesis and obesity development (miR-30b, miR-103, miR-148a, miR-17, miR-let-7a, miR-222, miR-378, miR-130a, miR-128, miR-27b, miR-34a, miR-484, miR-642a, miR-30c, miR-448, miR-302b, and miR-224).

As for miR-30b and miR-378, there is evidence that their overexpression can stimulate adipogenesis through participating in white fat adipogenesis and, finally, upregulating genes that are involved in the lipogenic process. The MiR-let7 family includes abundant breast milk microRNAs, such as miR-let-7a, miR-let-7b, miR-let-7c, and miR-let-7g, and plays an important role in the regulation of adipogenesis in human and mouse models. Additionally, miR-let-7a, miR-let-7b, and miR-let-7c found in breast milk are putative targets of leptin and adiponectin receptors.

Another abundant miRNA in breast milk is miR-148a. Recent studies have shown that an increase in exposure after ingestion of miR-148a will result in late infant development and growth and fat acquisition through the decreased action of insulin. MiR-148a also targets genes involved in and important to adipogenesis, insulin signaling, and the metabolism of energy. Results of the same study showed that decreased levels of miR-148a may protect infants of OW/OB mothers from obesity development [49].

MiR-130a-3p, miR-27b-3p, and miR-34a-5p have been proven to target PPARγ in vitro. PPARγ is a receptor of the nucleus, initially observed in adipose tissue and already known to participate in the regulation of adipogenesis in mammals. PPARγ was identified as a central transcriptional regulator of genes associated with triglyceride synthesis and lipid accumulation [5. In addition, miR-642a, miR-30c, miR-448, and miR-302b were found to be associated with adipogenic pathways, suggesting potential roles in modulating lipid metabolism and providing protection against adverse metabolic outcomes in infants.. Specifically, the miR-30 family in general has been observed to contribute to adipocyte differentiation. Previous studies showed that inhibition of the miR-30 family stopped adipogenesis, while overexpression resulted in stimulation of the process [55].

Mir-484 plays an important role in glucose synthesis and lipid metabolism. A previous study showed that miR-484 was significantly upregulated appropriately for gestational age (AGA) and born small for gestational age (SGA) obese children.

#### 3.3.2. Main Results Associated with the Most Abundant miRNAs

According to the included studies, miR-30a, miR-146b, miR-148a, miR-30b, miR-let-7a, miR-17, miR-103, miR-222, miR-let-7c, miR-let-7b, miR-let-7g, miR-200c, and miR-378 are among the most abundant human milk miRNAs in general that also contribute to obesity development or adipogenesis in infants. The most discussed miRNAs among the studies included are miR-30b, miR-let-7a, and miR-148a. Table 4 presents the associations between these three miRNAs and weight gain or obesity-related factors that were described in the included studies.

## 4. Discussion

Several miRNAs that participate in adipogenesis and adipocyte differentiation are expressed in the human milk of overweight or obese mothers and, thus, transfer the genetic predisposition to offspring. In summary, some of them (miR-30b and miR-let-7a) are positively associated with maternal adiposity or weight gain during pregnancy, whereas others do not show an association (miR-148a), or even show an inverse association with, for instance, maternal pre-pregnancy BMI (miR-30b and miR-let-7a), as shown in Table 4.

A factor that influences the outcomes of each study is the method of miRNA analysis used. Real-time PCR, a commonly used technique for targeted analysis, is very sensitive and specific in identifying low levels of miRNAs; however, it has the limitation of quantifying only a restricted number of miRNAs, which are preselected by the researchers according to a hypothesis-based approach. On the other hand, miRNA sequencing methods, like miRNA WTA (whole transcriptome assay) or small-RNA library preparation and RNA sequencing, offer comprehensive insights into the whole miRNAome, although it is less sensitive in identifying miRNAs that are expressed in low quantities. A single study utilized the Nanostring nCounter method of small RNA profiling. These panels use a type of technology for miRNA profiling without amplification of distinct RNAs or the need for reverse transcription.

Additionally, only four out of the seven studies took anthropometric measurements of the infants, and studied the correlations between miRNA expression, maternal BMI, and infant weight characteristics. In contrast, the other three studies correlated the abundance of the identified miRNAs in the body fluid studied with the maternal weight characteristics and made inferences about the possibility of obesity development in the offspring, based on the already published literature.

Based on the included articles (Table 3), miRNA consistency is dynamically modified through the lactation period. However, an overall abundance in the miRNAs described was observed in all studies. The major microRNA families that are expressed in breast milk and involved in obesity development are discussed below.

### 4.1. Interfering Factors

#### 4.1.1. Types of Samples

In the current review, six of the included studies used milk samples obtained from breastfeeding mothers at different lactating periods, while one used urine and plasma samples, in order to check the miRNA expression indirectly, from preterm infants after the consumption of breast milk.

There are numerous differences between breast milk and urine/plasma miRNAs. The former are produced by the mother’s mammary gland and secreted into milk, programmed to be transferred to the infant and participate in the regulation of processes like gene expression and metabolic programming. Their presence in human milk may be affected by maternal factors, such as nutrition, overall health status, or hormonal state. On the contrary, urine/plasma miRNAs are produced by infant cells, reflecting the health state of the infant that is influenced by its own environment, diet, and overall health. MiRNAs in these body fluids might be a result of cellular processes, tissue-specific expressions, or responses to internal and external stimuli.

Differentially expressed miRNAs in breast milk and infant urine/plasma have different actions and, therefore, may result in different outcomes. It might be possible that the miRNAs in breast milk primarily impact the infant’s early development and overall health by controlling the infant’s digestive system, immunity, and absorption of nutrients. On the other hand, miRNAs from infant urine or plasma could provide details regarding their internal state, including metabolic health, stress, or the presence of disease. The infant’s willingness to adjust to outside factors, like breast milk nutrition, may be represented in these miRNAs, which may not be directly modified by breast milk.

Maternal and infant health statuses play an important role in miRNA expression. For instance, it could be possible that maternal obesity affects breast milk miRNA content, eventually leading to altered metabolic outcomes in offspring; however, more studies are necessary to investigate this. Apart from that, the miRNA profile of an infant may be reflective of how the infant’s body is responding to breast milk consumption, along with other, potentially environmental, factors.

Finally, different conditions and miRNA origins may affect the differential expression that occurs in breast milk opposed to the baby’s urine or plasma. Recent studies suggest that breast milk miRNAs have been proposed to exert more direct influences on infant development, whereas miRNAs detected in plasma or urine more accurately reflect the infant’s internal physiological state and their responses to external and environmental factors, including breast milk consuming..

#### 4.1.2. Gestational Age

Another potentially important factor that may influence miRNA content in breast milk is gestational age. Term infants are born during the optimal period in line with the readiness of the infant’s immune system and growth. Usually, the miRNA profile is consistent with the healthy development of their immune and metabolic systems. These miRNAs could probably help to regulate the metabolism of fat, sensitivity to insulin, and balance of energy, which are necessary for the prevention of obesity during the growth of children [26].

Preterm infants differ in the period of delivery and, for this reason, their nutritional requirements are different. Mother’s milk seems to adapt to early infant needs, and so the consistency of miRNA in breast milk is variable [59]. Previous studies have indicated that human milk of pre-term infants contains increased amounts of miRNAs implicated in immune regulation and infant growth [60].

In conclusion, based on the recent literature and the current review, it can be assumed that a balanced miRNA profile in human milk might help improve the health status of infants and prevent them from eventually developing obesity. However, in premature infants, differences in miRNA setup, along with complications caused by premature delivery, may impact metabolic programming, raising their chance of developing obesity in later life. Therefore, prematurity might be a maternal factor influencing the miRNA content in human breast milk and, thus, the abundance of specific miRNAs that have been found. Consequently, understanding and enhancing the miRNA composition of breast milk might have important implications for decreasing the risk of obesity development in both full-term and premature infants.

### 4.2. The miR-30 Family

In this review, three members of the miR-30 family are most abundantly expressed in breast milk and are discussed below: miR-30a [48,54], miR-30b [49,50,51,52], and miR-30c [52]. Mir-30b is one of the most discussed in the included reviews, and among the most abundant miRNAs expressed in human breast milk. As mentioned in the previous literature, miR-30b is known to participate in the pathogenesis of obesity, promoting adipogenesis via adipocyte differentiation (white and brown adipose tissue development) and energy metabolism [26,55]. The fact that miR-30b was observed as being downregulated in OW/OB mothers compared to NW probably leads to an upregulation of its target genes. Furthermore, downregulating miR-30b during adipogenesis might have an impact on fat cell development, probably leading to a reduction in fat cell formation and potentially influencing the body’s ability to maintain energy balance and healthy fat tissue.

Additionally, as the maternal BMI increased before pregnancy, the expression levels of miR-30b decreased, which may indicate that women with higher BMIs tend to have decreased levels of miR-30b. This indicates a reverse relationship between maternal pre-pregnancy BMI and miR-30b, suggesting that women with a higher BMI prior to pregnancy generally exhibit lower levels of miR-30b. This downregulation probably suggests metabolic issues associated with obesity, specifically modifications in the formation of fat cells and insulin resistance. Changes like these may influence the regulation of miRNAs, such as miR-30b, which could contribute to the fat cell synthesis process.

On the other hand, the positive correlation between miR-30b and weight gain during pregnancy [50] may imply that changes in the body’s energy homeostasis and fat storage requirements influence miR-30b expression levels. Moreover, maternal pregnancy weight gain within healthy ranges is crucial for the growth of the fetus. Therefore, an increase in miR-30b expression may be part of the body’s adaptive response to control the expansion of fat tissue, while preserving the proper balance of energy.

The opposing results might indicate that miR-30b is regulated in different ways in response to pre-pregnancy BMI and the physiologic needs of pregnancy. Being overweight prior to pregnancy, commonly linked to metabolic stress and dysfunction, may cause, as an adverse effect, the inhibition of miR-30b. However, miR-30b increases may be in response to pregnancy’s increased biological requirements and weight gain. This rise in miR-30b may potentially support body fat cell creation and the storage of energy in a controlled way.

This reduction in miR-30b levels associated with elevated pre-pregnancy BMI might lead to altered adipose tissue function and metabolic errors, potentially transferring these effects to children, and probably increasing the possibility of developing obesity and other co-morbidities. On the contrary, the rise in miR-30b when taking into account weight gain during pregnancy might act as a protective mechanism in order to guarantee sufficient energy intake and enhance the growth of healthy fat tissue, a beneficial factor for both the mother and the developing baby.

The results presented here could imply that miR-30b may play a role in controlling how the organism reacts to different metabolic states. It appears to remain suppressed in high pre-pregnancy BMI but is triggered when gaining weight while pregnant in order to assist embryonic growth. The relationship between miR-30b concentrations, pre-pregnancy BMI, and maternal weight gain during pregnancy emphasizes the complex process of metabolic regulation through gestation. MiR-30b might function as a biological link between the metabolic state of the mother and the health effects on both the mother and her child. Gaining knowledge of these connections may help in developing strategies to improve the health of mothers and reduce the risk of metabolic issues in later generations.

As for the association of miR-30b levels and infant anthropometric measurements, a positive association was observed during early infancy. Finally, the outcomes may suggest that milk miR-30b may participate in ealry adipogenesisand thus may influence body composition of infants, with lower levels observed in milk from obese mothers possibly limiting this regulatory effect [49].

### 4.3. The Let-7 Family

Another important family of microRNAs with significant implications in obesity development in children is the miR-let7 family, including the following: miR-let-7a [48,49,50,52], miR-let-7b [48,52], miR-let-7c [48], and miR-let-7g [52]. Mir-let-7a is the most studied among the others of the family. It is abundantly expressed in human milk and implicated in lipid metabolism by contributing to the regulation of PUFA and SFA balance in human breast milk [48]. Based on the properties of fatty acids in milk, it may result in balancing the nutritional quality of human milk.

Its dynamic expression levels and, in particular, the decline it faces during lactation, might suggest its function in regulating the composition of milk in accordance with the increasing energy demands of the developing newborn. The presence of large amounts in colostrum during the early stages seems to be crucial for the rapid growth of infants, while a decline in mature milk probably assists the adjustment of the infant’s metabolism during the introduction of solids.

According to the included studies, the negative association between miR-let-7a and maternal BMI probably indicates that as BMI increases, the expression levels of this miRNA decrease in human milk. This may contribute to an adverse lipid composition in human milk, characterized by increased SFAs and decreased PUFAs. Changes like these might result in an adverse effect regarding the metabolic programming of the infant, possibly enhancing the risk of developing obesity.

Apart from that, the disturbed lipid profile may be implicated in changes in miR-let-7a levels, especially the increase in SFAs and reduction in PUFAs, which could result in larger deposits of fat and inflammation in children. Therefore, this increases the chance of obesity development, along with the associated metabolic diseases, as they age.

It was also observed that increased amounts of miR-let-7a in NW women during the initial months of breastfeeding might have a beneficial impact through encouraging the establishment of a healthy metabolic basis for the newborn. As miRNA levels decline over time in lactation, the progressive change in the level of lipids may help preserve a balanced caloric intake corresponding with the infant’s developmental phase, thus decreasing the chance of additional weight gain.

The presence of miR-let-7a in human milk appears to have a major effect on managing the lipid profile and impacting the metabolic health of newborns. Different concentrations of miR-let-7a, especially its correlations with maternal BMI and the content of fatty acids in breast milk, show that it might have an essential role in influencing the risk of an infant becoming obese. The breast milk of OW/OB mothers may include less miR-let-7a, which might result in an undesirable lipid profile for the child. This might elevate the probability of developing obesity or associated metabolic conditions in later years.

Another important aspect is the involvement of let-7 in targeting HMGA2, a protein that modifies the structure of chromatin in the nucleus, which illustrates how miRNAs can modulate chromatin-associated proteins to influence transcriptional accessibility and lineage progression. By repressing HMGA2, let-7 facilitates the shift from the proliferative clonal expansion phase to terminal adipocyte differentiation, thereby ensuring the proper timing and coordination of adipogenic events [58]. Conversely, miR-148a functions at an earlier stage by targeting DKK1, an inhibitor of the Wnt signaling pathway, effectively promoting the initiation of adipogenic commitment. These findings suggest that distinct miRNAs manage different stages of adipogenesis through both epigenetic configuration and signaling crosstalk. The complementary actions of miR-148a and let-7 illustrate a layered regulatory network in which miRNAs make small adjustments to balance between proliferation, differentiation, and chromatin remodeling. Understanding these mechanisms may provide valuable insight into the molecular basis of adipose tissue development and the dysregulation of adipogenesis observed in metabolic disorders such as obesity and insulin resistance.

### 4.4. Importance of miR-148a

MiR-148a is another microRNA molecule among the most studied and abundant in human milk that has been identified as participating in obesity development. As discussed in three of the included studies [48,49,52], a downregulation of miR-148a was observed in OW/OB mothers compared to NW [49]. Both previous studies [49,52] observed negative and significant associations between miR-148a and infant weight and body measurements, indicating the protective role of miR-148a in obesity and, consequently T2D, in offspring. The above findings are suggesting that maternal metabolic status may possibly influence infant metabolic programming by altering epigenetic and lipid-regulatory signaling pathways during the early period of life..

However, miR-148a in human milk is regulated by hormonal, circadian, and stress-related factors, which may be implicated in neonatal development and metabolic programming. Oxytocin is a neuropeptide important to lactation and maternal behavior, and has been shown to upregulate specific miRNAs, including miR-148a, thereby linking maternal neuroendocrine responses to the molecular composition of early milk [61]. This regulation suggests that cesarean delivery, which is associated with attenuated oxytocin release, may lead to reduced secretions of miR-148a in milk, potentially affecting the epigenetic and immunological signaling transferred to the neonate [35]. Moreover, melatonin, a circadian hormone known to modulate microRNA expression, also influences miR-148a-3p regulation, as demonstrated in breast-tissue models [62]. Given that most human milk sampling in the existing studies has occurred during the daytime, when melatonin levels are physiologically low, this temporal bias may contribute to underestimations of miR-148a expression and obscure its full circadian dynamics.

In addition to the above, maternal stress has been shown to suppress miR-148a expression in breast milk, suggesting impaired epithelial barrier function [35]. These findings collectively underscore the sensitivity of milk-derived miR-148a to the maternal hormonal environment.

### 4.5. Association Between MicroRNAs and Breast Milk Constituents

Zamanillo et al. observed a pattern in the constituents of milk from NW mothers, which was absent in OW/OB mothers, between the different lipid components (MUFAs, SFAs, and triglycerides) at various stages of lactation and the miRNAs in breast milk. The results suggest that these may be important for the development of infants. In general, MUFAs are related to improved metabolic results, so the negative correlation between MUFAs and breast milk miRNAs that was observed during the first month of lactation may be a way for the infant to achieve a more balanced and regulated intake of fatty acids during early life. SFAs are a type of fat that is richer in energy, which may be vital for the infant’s increased growth and development. During this period, the infant’s rising need for energy possibly triggers an increase in miRNA expression, which in turn promotes the content of saturated fats in breast milk. Triglycerides are an important source of energy in breast milk, and this link might point to an effort to satisfy the infant’s energy needs as they grow up.

The altered pattern observed in the case of OW/OB mothers may be linked to metabolic changes that are related to obesity. These changes might end up in a difference in the lipid profile of BM, which can be controlled by miRNA expression. The disturbance in miRNA expression can cause an uneven lipid profile in the milk, defined by variations in levels of MUFAs, SFAs, and triglycerides. Such an imbalance could possibly affect the metabolic growth of the infant, which could result in later complications. Infants’ metabolic programming can be altered by early exposure to breast milk, resulting in a different lipid profile. In particular, adiposity, inflammation, and fat formation may be enhanced in a diet rich in SFAs but low in MUFAs, which in turn might increase the risk of obesity and other associated metabolic diseases in future generations.

The identified associations between microRNAs and lipid compounds in the breast milk of NW mothers are expected to have an important effect on regulating the growth and metabolism of infants. Yet, in mothers who are OW or OB, abnormalities in these patterns can potentially lead to obesity in their children through modifying the dietary content of breast milk and impacting early metabolic programming. This emphasizes the significance of preserving maternal health and ensuring an adequate diet during breastfeeding in order to promote the healthy growth of infants.

### 4.6. Breast Milk miRNAs and the Wnt-FTO Metabolic Axis

Activation of canonical Wnt/β-catenin signaling suppresses the induction of PPARγ/C/EBP-driven pathways, while attenuation of Wnt signaling allows adipocyte commitment [21,22]. Overlapping in this canonical control, epigenetic and epitranscriptional regulators regulate both the expression of the components of the Wnt pathway and the biosynthesis and stability of small non-coding RNAs involved in lineage decisions. In this context, the three most significantly expressed breast milk miRNAs—miR-148a, let-7a, and miR-30b—occupy distinct but intersecting hubs: miR-148a, abundant in human milk, represses DNMTs and, thus, may reduce promoter methylation of Wnt pathway genes. This way, sustained Wnt tone that opposes premature adipogenesis is favored [16,26]. In contrast, the reduction in miR-148a levels reported in formula and in some groups of OW/OB mothers [35,36,49] would be expected to relieve DNMT suppression, increase Wnt promoter methylation, and reduce anti-lipogenic Wnt signaling, thereby lowering the threshold for adipogenic differentiation.

Members of the let-7 family, including let-7a, regulate cell proliferation and have been associated in milk with maternal metabolic status [48,50]. Altering let-7a abundance might, therefore, alter the proliferative versus differentiative balance of progenitors and interact with the Wnt signaling pathway. MiR-30b has been implicated in lipogenic regulatory networks and may influence adipocyte maturation downstream of Wnt attenuation.

Importantly, FTO-dependent m^6^A demethylation provides an orthogonal regulatory layer by altering mRNA stability, splicing, and translation of lipogenic regulators (e.g., SREBP1c and CIDEC), and likely influencing the biogenesis or turnover of miRNA precursors through m^6^A-sensitive processing steps [25,32,33]. Thus, increased FTO activity in target progenitors would enhance the expression of lipogenic genes, even when Wnt signaling is partially active, and in combination with reduced anti-DNMT miRNAs (e.g., lower miR-148a in formula or maternal obesity) would cause adipogenic commitment.

Epidemiological observations linking maternal overweight/obesity to altered milk miRNA profiles and to early infant adiposity are, therefore, compatible with a model in which milk miRNA abundance, Wnt pathway methylation status, and FTO-mediated m^6^A dynamics lead to adipocyte lineage outcomes. Experimental testing of this interacting triad will require simultaneous measurements of milk miRNAs, DNMT activity/Wnt promoter methylation, and cellular m^6^A landscapes in human-related progenitor models.

### 4.7. Future Perspectives

Further studies are necessary to support the evidence linking miRNA expression levels to the development of obesity in offspring, based on the included data. In future studies, it may be useful to exclusively include breastfeeding infants, and take infant body measurements in order to compare infant growth. Additionally, other parameters that are involved in maternal lifestyle should be avoided, like smoking or alcohol consumption, due to the fact that they are environmental factors that interfere with miRNA expression. Future research should implement longitudinal and circadian-controlled sampling designs, consider the mode of delivery and maternal stress status, and explore oxytocin or melatonin modulation as potential interventions to sustain optimal miR-148a levels in lactation. Such approaches could clarify the role of miR-148a as a mediator linking maternal physiology with infant developmental outcomes. In addition, mothers with underlying conditions, such as Type 1 or 2 diabetes or other metabolic conditions, which could predispose obesity develoment, should be excluded. Another important factor to take into account is the BMI or the weight gain during pregnancy, and not only the pre-gestational BMI, because rapid weight gain or obesity during pregnancy is important for the miRNA expression levels in breast milk. Finally, future studies should include larger cohorts, and randomized controlled trials are needed to enhance the accuracy and reliability of findings.

## 5. Conclusions

This review underlines how extraordinary the ability of the maternal body is in taking care of another human body through breastfeeding—in particular, how breast milk production adapts to the infant’s needs and finally protects them from adverse consequences. Research has proposed, so far, that although some miRNAs found in breast milk are involved in the transmission of genetic predisposition, others provide a protective role against obesity in offspring. Finally, even though many different sets of miRNAs may be involved in infant development, the miRNA profile remains stable. These highlight the significance of maternal balanced nutrition and health during the lactation period. Nevertheless, further studies are needed to clarify the consequences of maternal overweight or obesity on human miRNAs and to identify how it is affecting the next generations. Apart from diet, it would be interesting to determine how other maternal lifestyle factors affect lactating offspring, such as exercise, during pregnancy or lactation.

## Figures and Tables

**Figure 1 genes-16-01373-f001:**
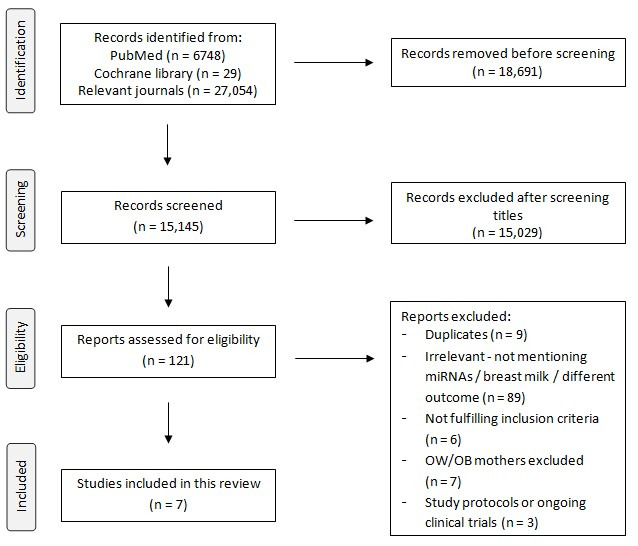
PRISMA flow diagram of the study selection process.

**Table 2 genes-16-01373-t002:** Included studies’ characteristics.

Author and Publication Date	Type of Study	Sample Characteristics	Method of miRNA Analysis	Maternal Characteristics	Infant Characteristics	
				Health Status BMI (kg/m^2^) Health Issues: GDM, Preeclampsia, T1D, Hypertension, etc. (%)	Breastfeeding Status (%)	Gestational Age (wks) Gender (Female, %)
Zamanillo R., 2019 [48]	Cohort study	• 59 lactating women (25–42 years old) • Milk (1, 2, and 3 mo)	RT-qPCR	• Mean BMI: pre-gestational BMI, 1, 2, 3 mo: All: 24.07 NW: 21.43 OW/OB: 28.87	• 86%—1 mo • 85%—2 mo • 78%—3 mo	• Gestational age: 39.1 ± 1.73 • Gender: 53
Shah K.B., 2021 [49]	Cohort study	• 365 mother–infant pairs (a subset of 60: 30OW/OB + 30NW) • Milk (1 and 3 mo)	Real-time PCR	• Pre-gestational BMI: NW: 22 ± 1.8 OW: 31.6 ± 7.3 • GDM, smoking and alcohol consumption excluded	• NW: 100%—1 mo 100%—3 mo 63.33%—6 mo • OW: 100%—1 mo 90.91%—3 mo 60%—6 mo	• Gestational age: NW: 39.26 ± 0.4 OW: 38.64 ± 0.9 • Gender: NW: 47 OW: 47
Xi Y., 2015 [50]	Observational study	• A total of 119 lactating mothers, 27–32 years old: • Group 1. 86 mothers donated colostrum (47 normal, 19 GDM, 20 GHD) • Group 2. 33 normal mothers donated mature milk • Collection at 2–5 days and 3 mo post-partum	Real-time PCR	• Pre-gestational BMI: 22.49 ± 3.89 • BMI late in pregnancy: 28.34 ± 3.57 • GDM: 22.1 • GHD: 23.3	100%—3 mo	• Gestational age: 39 • Gender: 50
Kupsco A., 2021 [51]	Cohort study	• 364 mother–infant pairs (24–35 years old) • Milk (1–7, 7–14, 14–30, >30 days post-delivery)	MiRNA WTA (sequencing)	• Pre-gestational BMI: 23.9 ± 3.85 • GDM: 9.1 • Preeclamptic: 1.6 • T1D: 0.5 • Smoking: 26.6	N/A	• Gestational age: 39.5 ± 1.37 • Gender: 48.4
Eun Y. Cho, 2022 [52]	Observational study	• 65 lactating women, ≥18 years old (47NW, 18OW/OB) • Milk (collected 66 days post-partum)	Nanostring nCounter method	• Pre-gestational BMI All: 25.2 ± 5.6 NW: 22.0± 1.9 OW/OB: 33.7 ± 2.5 • Current BMI All: 26.8 ± 5.1 NW: 24.1 ± 2.8 OW/OB: 33.8 ± 2.4 • GDM All: 4.6 NW: 2.2 OW/OB: 11.1 • Preeclampsia All: 7.7 NW: 2.2 OW/OB: 22.2	• Direct BF All: 87.7 NW: 91.5 OW/OB: 77.8 • Bottled feeding All: 53.8 NW: 46.8 OW/OB: 72.2 • Formula feeding All: 15.4 NW: 12.8 OW/OB: 22.2	• Gestational age: All: 39.1 ± 1.2 NW: 39 ± 1.3 OW/OB: 39.1 ± 0.8
Kim E.-B., 2024 [53]	Cohort study	• 33 preterm infants (12 breast-fed, 21 formula-fed) • Urine and serum of the infants during stay in the NICU	Small RNA library preparation; small RNA sequencing	• BMI Breast-fed: 29.22 ± 3.3 Formula-fed: 28.55 ± 2.68 • GDM Breast-fed: 41.67 Formula-fed: 28.57 • Hypertension Breast-fed: 50 Formula-fed: 61.9	2 groups of study, breast-fed and formula-fed	• Gestational age: Breast-fed: 29 + 0.86 ± 3.39 Formula-fed: 30.65 ± 2.55 • Gender: Breast-fed: 50 Formula-fed: 42.86
Van Syoc E., 2024 [54]	Cohort study	• 163 mother–infant pairs • Μilk (at birth, 1, and 4 mo)	Small RNA library preparation; small RNA sequencing	• Normal pre-gestational BMI: 38% • GDM: 11	69%—6 mo	Term infants: 37–42

**Table 3 genes-16-01373-t003:** MiRNAs in EVs related to obesity and adipogenesis in breastfeeding offspring, taking into account maternal BMI.

miRNAs	Role in Obesity Development
miR-17, miR-146b	Putative target of LEP and associated with obesity development and adipogenesis [48]
miR-27b, miR-34a, miR-128, miR-130a	Targets PPARγ in mice and can inhibit adipogenesis in vitro [51]
miR-30a	Target of LEP, LEPR, ADIPOR1, ADIPOR2 [48] Key regulatory role in human adipogenesis [55]
miR-30b	Exposure to overexpressed miR-30b may act along with fat mass increase in infants [49] Role in pathogenesis of adipogenesis, obesity, and metabolism [50] Key regulatory role in human adipogenesis [55]
miR-30c	Correlation to obesity and adipogenesis by disturbing the infant’s metabolism or protecting them from harmful results [52] Key regulatory role in human adipogenesis [55]
miR-103	Associated with obesity development and adipogenesis [48]
miR-148a	Target genes are involved in important pathways related to energy metabolism, insulin signaling, and adipogenesis [49]
miR-200c	Promotes adipogenesis in mouse models [56]
miR-222	Target of ADIPOR1, ADIPOR2, and LEPR [48]
miR-224	Participates in differentiation of adipocytes and metabolism of fatty acids [54]
miR-378	Role in pathogenesis of adipogenesis, obesity, and metabolism [50]
miR-642a, miR-448, miR-302b	Correlation to obesity and adipogenesis by disturbing the infant’s metabolism or protecting them from harmful results [52]
miR-484	Regulated glucolipid metabolism Potential regulator of insulin gene expression [57]
miR-let-7a	Target of ADIPOR1, ADIPOR2, and LEPR [48] Regulation of adipogenesis and obesity and important role in metabolism in both mice and humans [50]
miR-let-7b, miR-let-7c	Target of ADIPOR1, ADIPOR2, and LEPR [48] Important regulation of adipogenesis [58]
miR-let-7g	Important regulation of adipogenesis [58]

**Table 4 genes-16-01373-t004:** Most abundant miRNAs found in breast milk and their association with weight gain or obesity-related factors.

Most Abundant miRNAs	Significant Association	Positive Association	Negative Association	No Association	Downregulation
**miR-30b**	Infant body measurements	Maternal weight gain during pregnancy	Maternal pre-pregnancy BMI, maternal BMI late in pregnancy	-	Colostrum of OW/OB mothers
**miR-let-7a**	-	Maternal weight gain during pregnancy	Milk adiponectin in NW women, maternal weight late in pregnancy, maternal pre-pregnancy BMI, BMI late in pregnancy	-	Colostrum and mature milk in OW/OB mothers
**miR-148a**	-	-	Milk adiponectin in NW women, infant body measurements of NW women	OW/OB mothers	-

## Data Availability

This review article does not report new data. All data discussed are from previously published studies, which are appropriately cited.

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
