# Peer review of "Human Breast Milk miRNAs: Investigation of Association Between Breastfeeding Children and Maternal Obesity in Obesity Development in Offspring"

_genes, 2025, doi:10.3390/genes16111373_

Round 1

Reviewer 1 Report

Comments and Suggestions for Authors

The authors have undertaken to describe an interesting topic, the work was developed using Prisma software.
The introduction would have included a few sentences about the components of human milk, why it is so important, what properties it has, and which components, besides mRNA, may influence childhood obesity.
The work often lacks citations; please carefully review, for example, the two paragraphs in subsection 1.2.
In my opinion, Table 2 needs to be revised; it's difficult to read when it's split across two pages. Is the title of the work necessary? We have the author and the year of publication; in my opinion, the title is unnecessary and takes up a lot of space in the table. Perhaps combining "gestational age" and "gender" would be a good idea? The table needs to be reworded.
Table 3 - needs to be standardized; the beginning of the table is centered, and the rest of the text is on the left side.
For a review, the number of articles used is very small (only 33). I understand the authors' intention regarding the inclusion and exclusion criteria and the scarcity of research on this topic. However, 33 cited articles, including the introduction, seems quite small.

Reviewer 2 Report

Comments and Suggestions for Authors

Review of manuscript ID: genes-3903702

Human breast milk miRNAs: Investigation of association between breastfeeding children and maternal obesity in obesity development in the offspring.

Marina Chondrogianni, Maria Lithoxopoulou, Athina Ververi, Alexandros Lampropoulos, Alexandros Sotiriadis, Eystratios Kolibianakis

General comments:

It is the aim of this systematic review to explore the hypothesis that miRNA (miR) content in breast milk might be influenced by maternal obesity, eventually affecting obesity risk in the offspring. Among the most consistently reported miRs in the literature are miR-148a, the miR-30 family, and the miR-let-7 family - particularly miR-30b and miR-let-7a - which are involved in the regulation of adipogenesis.

This authors, who are new in the field, collected translational evidence for the involvement of miRs in the regulation of adipogenesis but miss important recently recognized and published aspects that should be included to provide a more balanced and updated view on this topic.

Specific comments:

Introduction

The introduction misses recent evidence that milk exosomal miRs are taken up by the infant´s intestine and reach the systemic circulation (vertical transmission).

The “functional hypothesis” of milk exosomal miR transfer opposed to the “nutritional hypothesis” (milk exosomes may only serve as simple nutrients) should be mentioned in the introduction.

Melnik BC, Kakulas F, Geddes DT, Hartmann PE, John SM, Carrera-Bastos P, Cordain L, Schmitz G. Milk miRNAs: simple nutrients or systemic functional regulators? Nutr Metab (Lond). 2016 Jun 21;13:42. doi: 10.1186/s12986-016-0101-2. PMID: 27330539; PMCID: PMC4915038.

The Zempleni group provided evidence that labelled bovine milk exosomes reached various distant organs of mice after oral exosome gavage.

Zempleni J, Aguilar-Lozano A, Sadri M, Sukreet S, Manca S, Wu D, Zhou F, Mutai E. Biological Activities of Extracellular Vesicles and Their Cargos from Bovine and Human Milk in Humans and Implications for Infants. J Nutr. 2017 Jan;147(1):3-10. doi: 10.3945/jn.116.238949. Epub 2016 Nov 16. PMID: 27852870; PMCID: PMC5177735.

Weil et al. demonstrated intestinal epithelial cellular miR uptake, and AGO2 loading of milk miRs in neonates using xenomiRs as tracers.

Weil PP, Reincke S, Hirsch CA, Giachero F, Aydin M, Scholz J, Jönsson F, Hagedorn C, Nguyen DN, Thymann T, Pembaur A, Orth V, Wünsche V, Jiang PP, Wirth S, Jenke ACW, Sangild PT, Kreppel F, Postberg J. Uncovering the gastrointestinal passage, intestinal epithelial cellular uptake, and AGO2 loading of milk miRNAs in neonates using xenomiRs as tracers. Am J Clin Nutr. 2023 Jun;117(6):1195-1210. doi: 10.1016/j.ajcnut.2023.03.016. Epub 2023 Mar 22. PMID: 36963568.

Recently, Yung et al. provided evidence that neonatal human enteroids rapidly

take up digested extracelluar vesicles (dEVs) in part via clathrin-mediated endocytosis. 

Yung C, Zhang Y, Kuhn M, Armstrong RJ, Olyaei A, Aloia M, Scottoline B, Andres SF. Neonatal enteroids absorb extracellular vesicles from human milk-fed infant digestive fluid. J Extracell Vesicles. 2024 Apr;13(4):e12422. doi: 10.1002/jev2.12422. PMID: 38602306; PMCID: PMC11007820.

Furthermore, milk exosomes have an impact on the colonization of Bifidobacteria. They are taken up by Bifidobacteria and promote their growth.

Luo Y, Bi J, Lin Y, He J, Wu S, Zhang Y, Wang Y, Song S, Guo H. Milk-derived small extracellular vesicles promote bifidobacteria growth by accelerating carbohydrate metabolism. LWT, 2023;182:114866, doi: 10.1016/j.lwt.2023.114866.

Cristóbal-Cañadas D, Parrón-Carrillo R, Parrón-Carreño T. Exosomes in Breast Milk: Their Impact on the Intestinal Microbiota of the Newborn and Therapeutic Perspectives for High-Risk Neonates. Int J Mol Sci. 2025 Apr 5;26(7):3421. doi: 10.3390/ijms26073421. PMID: 40244312; PMCID: PMC11989396.

miR-148a

The authors should acknowledge that miR-148a-3p is induced by oxytocin, the key hormone of physiological vaginal delivery.

Gutman-Ido E, Reif S, Musseri M, Schabes T, Golan-Gerstl R. Oxytocin Regulates the Expression of Selected Colostrum-derived microRNAs. J Pediatr Gastroenterol Nutr. 2022 Jan 1;74(1):e8-e15. doi: 10.1097/MPG.0000000000003277. PMID: 34371509.

Compared to vaginal delivery, breastmilk miR-148a levels are decreased in mothers who delivered by Cesarean section (>50% in Greece).

Chiba T, Kooka A, Kowatari K, Yoshizawa M, Chiba N, Takaguri A, Fukushi Y, Hongo F, Sato H, Wada S. Expression profiles of hsa-miR-148a-3p and hsa-miR-125b-5p in human breast milk and infant formulae. Int Breastfeed J. 2022 Jan 3;17(1):1. doi: 10.1186/s13006-021-00436-7. PMID: 34980190; PMCID: PMC8725387.

Melatonin also controls the expression of miR-148a-3p.

Lacerda JZ, Ferreira LC, Lopes BC, Aristizábal-Pachón AF, Bajgelman MC, Borin TF, Zuccari DAPC. Therapeutic Potential of Melatonin in the Regulation of MiR-148a-3p and Angiogenic Factors in Breast Cancer. Microrna. 2019;8(3):237-247. doi: 10.2174/2211536608666190219095426. PMID: 30806335.

Most breast milk miR analyses have been performed during day time of hospital staff working hours. Thus, there is a significant gap of knowledge of miR-148a expression during night time breast feeding periods and maternal sleep characteristics.

Maternal stress suppresses milk-derived miR-148a.

Chiba T, Takaguri A, Kooka A, Kowatari K, Yoshizawa M, Fukushi Y, Hongo F, Sato H, Fujisawa M, Wada S, Maeda T. Suppression of milk-derived miR-148a caused by stress plays a role in the decrease in intestinal ZO-1 expression in infants. Clin Nutr. 2022 Dec;41(12):2691-2698. doi: 10.1016/j.clnu.2022.10.004. Epub 2022 Oct 13. PMID: 36343560.

In the early steps of adipogenesis, especially during adipocyte stem cell development, Wnt signaling is most criticaly involved.

Prestwich TC, Macdougald OA. Wnt/beta-catenin signaling in adipogenesis and metabolism. Curr Opin Cell Biol. 2007 Dec;19(6):612-7. doi: 10.1016/j.ceb.2007.09.014. Epub 2007 Nov 9. PMID: 17997088; PMCID: PMC2709272.

Christodoulides C, Lagathu C, Sethi JK, Vidal-Puig A. Adipogenesis and WNT signalling. Trends Endocrinol Metab. 2009 Jan;20(1):16-24. doi: 10.1016/j.tem.2008.09.002. Epub 2008 Nov 13. PMID: 19008118; PMCID: PMC4304002.

D'Alimonte I, Lannutti A, Pipino C, Di Tomo P, Pierdomenico L, Cianci E, Antonucci I, Marchisio M, Romano M, Stuppia L, Caciagli F, Pandolfi A, Ciccarelli R. Wnt signaling behaves as a "master regulator" in the osteogenic and adipogenic commitment of human amniotic fluid mesenchymal stem cells. Stem Cell Rev Rep. 2013 Oct;9(5):642-54. doi: 10.1007/s12015-013-9436-5. PMID: 23605563; PMCID: PMC3785124.

Bagchi DP, MacDougald OA. Wnt Signaling: From Mesenchymal Cell Fate to Lipogenesis and Other Mature Adipocyte Functions. Diabetes. 2021 Jul;70(7):1419-1430. doi: 10.2337/dbi20-0015. Epub 2021 Jun 21. PMID: 34155042; PMCID: PMC8336005.

Decreased Wnt signaling enhances adipocyte stem cell development.

Umbilcal cord mesenchymal stem cells of obese mothers exhibit decreased Wnt signaling.

Boyle KE, Patinkin ZW, Shapiro AL, Baker PR 2nd, Dabelea D, Friedman JE. Mesenchymal Stem Cells From Infants Born to Obese Mothers Exhibit Greater Potential for Adipogenesis: The Healthy Start BabyBUMP Project. Diabetes. 2016 Mar;65(3):647-59. doi: 10.2337/db15-0849. Epub 2015 Dec 2. PMID: 26631736; PMCID: PMC4764150.

Wnt signaling is controlled by various Wnt inhibitors, especially Dickkopf 1 (DKK1).

miR-148a-3p is overexpressed in milk exosomes of preterm infants.

Kahn S, Liao Y, Du X, Xu W, Li J, Lönnerdal B. Exosomal MicroRNAs in Milk from Mothers Delivering Preterm Infants Survive in Vitro Digestion and Are Taken Up by Human Intestinal Cells. Mol Nutr Food Res. 2018 Jun;62(11):e1701050. doi: 10.1002/mnfr.201701050. Epub 2018 May 18. PMID: 29644801.

Ma L, Huo Y, Tang Q, Wang X, Wang W, Wu D, Li Y, Chen L, Wang S, Zhu Y, Wang W, Liu Y, Xu N, Chen L, Yu G, Chen J. Human Breast Milk Exosomal miRNAs are Influenced by Premature Delivery and Affect Neurodevelopment. Mol Nutr Food Res. 2024 May;68(9):e2300113. doi: 10.1002/mnfr.202300113. Epub 2024 Apr 21. PMID: 38644336.

miR-148a-3p targets the Wnt inhibitor DKK1.

Sheng W, Jiang H, Yuan H, Li S. miR-148a-3p facilitates osteogenic differentiation of fibroblasts in ankylosing spondylitis by activating the Wnt pathway and targeting DKK1. Exp Ther Med. 2022 May;23(5):365. doi: 10.3892/etm.2022.11292. Epub 2022 Apr 1. PMID: 35493425; PMCID: PMC9019766.

miR-22-3p, another highly expressed exosomal miR of premature breast milk, targets FOSB, a transcriptional inducer of DKK1.

For further details of milk exosomal miR-mediated regulation of adipogenesis and islet beta-cell homeostasis see

Melnik BC, Weiskirchen R, John SM, Stremmel W, Leitzmann C, Weiskirchen S, Schmitz G. White Adipocyte Stem Cell Expansion Through Infant Formula Feeding: New Insights into Epigenetic Programming Explaining the Early Protein Hypothesis of Obesity. Int J Mol Sci. 2025 May 8;26(10):4493. doi: 10.3390/ijms26104493. PMID: 40429638; PMCID: PMC12110815.

Melnik BC, Weiskirchen R, Weiskirchen S, Stremmel W, John SM, Leitzmann C, Schmitz G. Diabetes-preventive molecular mechanisms of breast versus formula feeding: new insights into the impact of milk on stem cell Wnt signaling. Front Nutr. 2025 Jul 29;12:1652297. doi: 10.3389/fnut.2025.1652297. PMID: 40799516; PMCID: PMC12341479.

Translational evidence indicates that miR-148a-3p targets ZFP217, which is involved in the expression of fat mass and obesity associated gene (FTO), which via m6A demethylation plays a key role in adipogenesis.

Yang Z, Yu GL, Zhu X, Peng TH, Lv YC. Critical roles of FTO-mediated mRNA m6A demethylation in regulating adipogenesis and lipid metabolism: Implications in lipid metabolic disorders. Genes Dis. 2021 Jan 28;9(1):51-61. doi: 10.1016/j.gendis.2021.01.005. PMID: 35005107; PMCID: PMC8720706.

In addition, milk exosomal miR-148a-3p targets DNA methyltransferase 1 (DNMT1).

Melnik BC, Schmitz G. MicroRNAs: Milk's epigenetic regulators. Best Pract Res Clin Endocrinol Metab. 2017 Aug;31(4):427-442. doi: 10.1016/j.beem.2017.10.003. Epub 2017 Oct 20. PMID: 29221571.

Golan-Gerstl R, Elbaum Shiff Y, Moshayoff V, Schecter D, Leshkowitz D, Reif S. Characterization and biological function of milk-derived miRNAs. Mol Nutr Food Res. 2017 Oct;61(10). doi: 10.1002/mnfr.201700009. Epub 2017 Jul 31. PMID: 28643865.

Reif S, Elbaum Shiff Y, Golan-Gerstl R. Milk-derived exosomes (MDEs) have a different biological effect on normal fetal colon epithelial cells compared to colon tumor cells in a miRNA-dependent manner. J Transl Med. 2019 Sep 30;17(1):325. doi: 10.1186/s12967-019-2072-3. PMID: 31564251; PMCID: PMC6767636.

DNMT1 also regulates Wnts gene expression.

The absence of milk exosomal miRNAs (miRNA-148a-3p, miRNA-22-3p) in formula, that physiologically target DNA methyltransferase 1 and 3A (DNMTs), may increase WNT promoter methylation, further suppressing WNT gene expression, another adipogenic constellation.

These epigenetic aspects are of key importance to understand adipogenesis and have to be presented in more detail.

Furthermore, this review completely ignores the role of FTO, which controls m6A mRNAs, a further important regulatory layer affecting gene expression.

FTO is involved in the regulation of adipocyte stem cell homeostasis.

Martin Carli JF, LeDuc CA, Zhang Y, Stratigopoulos G, Leibel RL. FTO mediates cell-autonomous effects on adipogenesis and adipocyte lipid content by regulating gene expression via 6mA DNA modifications. J Lipid Res. 2018 Aug;59(8):1446-1460. doi: 10.1194/jlr.M085555. Epub 2018 Jun 22. PMID: 29934339; PMCID: PMC6071765.

Melnik BC, Weiskirchen R, Stremmel W, John SM, Schmitz G. Risk of Fat Mass- and Obesity-Associated Gene-Dependent Obesogenic Programming by Formula Feeding Compared to Breastfeeding. Nutrients. 2024 Jul 28;16(15):2451. doi: 10.3390/nu16152451. PMID: 39125332; PMCID: PMC11314333.

FTO promotes SREBP1c maturation, a key transcription factor of lipogenesis.

Chen A, Chen X, Cheng S, Shu L, Yan M, Yao L, Wang B, Huang S, Zhou L, Yang Z, Liu G. FTO promotes SREBP1c maturation and enhances CIDEC transcription during lipid accumulation in HepG2 cells. Biochim Biophys Acta Mol Cell Biol Lipids. 2018 May;1863(5):538-548. doi: 10.1016/j.bbalip.2018.02.003. Epub 2018 Feb 25. PMID: 29486327.

Thus, miR-148a-3p is primarily an anti-adipogenic miR, and its levels are decreased in breast milk exosomes of mothers with obesity or gestational diabetes as shown by Shah et al. apparently an unfavorable epigenetic constellation promoting the development of obesity and diabetes of the offspring.

Infant formula does only contain trace amounts of miR-148a and fails completely in adequate physiological milk exosomal miR signaling.

Leiferman A, Shu J, Upadhyaya B, Cui J, Zempleni J. Storage of Extracellular Vesicles in Human Milk, and MicroRNA Profiles in Human Milk Exosomes and Infant Formulas. J Pediatr Gastroenterol Nutr. 2019 Aug;69(2):235-238. doi: 10.1097/MPG.0000000000002363. PMID: 31169664; PMCID: PMC6658346.

Chiba T, Kooka A, Kowatari K, Yoshizawa M, Chiba N, Takaguri A, Fukushi Y, Hongo F, Sato H, Wada S. Expression profiles of hsa-miR-148a-3p and hsa-miR-125b-5p in human breast milk and infant formulae. Int Breastfeed J. 2022 Jan 3;17(1):1. doi: 10.1186/s13006-021-00436-7. PMID: 34980190; PMCID: PMC8725387.

Cemali Ö, Çelik E, Deveci G, HirfanoÄŸlu İM, Önal EE, AÄŸagündüz D. Detection and quantification of miRNA 148a expression in infant formulas. J Food Sci. 2025 Jan;90(1):e17648. doi: 10.1111/1750-3841.17648. PMID: 39828407.

Noteworthy to mention that miR-148a, which via exosome uptake by Bifidobacteria targets critical genes involved in Bifidobacteria carbohydrate metabolisms promoting their growth.

Luo Y, Bi J, Lin Y, He J, Wu S, Zhang Y, Wang Y, Song S, Guo H. Milk-derived small extracellular vesicles promote bifidobacteria growth by accelerating carbohydrate metabolism. LWT, 2023;182:114866, doi: 10.1016/j.lwt.2023.114866.

Let-7

As shown in 3T3-L1 pre-adipocytes, let-7 plays an important role in adipocyte differentiation in part by targeting high mobility group A 2 (HMGA2), a protein that modulates the structure of chromatin in the nucleus. HMGA2, thereby regulates the transition from clonal expansion to terminal differentiation.

Sun T, Fu M, Bookout AL, Kliewer SA, Mangelsdorf DJ. MicroRNA let-7 regulates 3T3-L1 adipogenesis. Mol Endocrinol. 2009 Jun;23(6):925-31. doi: 10.1210/me.2008-0298. Epub 2009 Mar 26. Erratum in: Mol Endocrinol. 2009 Sep;23(9):1522. PMID: 19324969; PMCID: PMC2691679.

Thus, miR-148a via targeting the Wnt inhibitor DKK1 and let-7 miRs via targeting HMGA2 suppress very early and later steps of adipogenesis.

MiR-30b

Long-chain omega 3 fatty acids upregulate the expression of miR-30b.

Kim J, Okla M, Erickson A, Carr T, Natarajan SK, Chung S. Eicosapentaenoic Acid Potentiates Brown Thermogenesis through FFAR4-dependent Up-regulation of miR-30b and miR-378. J Biol Chem. 2016 Sep 23;291(39):20551-62. doi: 10.1074/jbc.M116.721480. Epub 2016 Aug 3. PMID: 27489163; PMCID: PMC5034049.

miR-30b is critical for the generation of brown adipose tissue and thermogenesis, an important mechanism controlling energy balance and the ratio of brown versus white adipose tissue distribution,

Hu F, Wang M, Xiao T, Yin B, He L, Meng W, Dong M, Liu F. miR-30 promotes thermogenesis and the development of beige fat by targeting RIP140. Diabetes. 2015 Jun;64(6):2056-68. doi: 10.2337/db14-1117. Epub 2015 Jan 9. PMID: 25576051; PMCID: PMC4876748.

Unfortunately, Shah et al. have not differentiated BAT vs. WAT development in their studies.

The role of milk exosomal miR-30b has already been discussed in the context of postnatal thermogenesis earlier.

Melnik BC, Stremmel W, Weiskirchen R, John SM, Schmitz G. Exosome-Derived MicroRNAs of Human Milk and Their Effects on Infant Health and Development. Biomolecules. 2021 Jun 7;11(6):851. doi: 10.3390/biom11060851. PMID: 34200323; PMCID: PMC8228670.

Importantly, FTO is a direct target of miR-30b.

Sun L, Gao M, Qian Q, Guo Z, Zhu P, Wang X, Wang H. Triclosan-induced abnormal expression of miR-30b regulates fto-mediated m6A methylation level to cause lipid metabolism disorder in zebrafish. Sci Total Environ. 2021 May 20;770:145285. doi: 10.1016/j.scitotenv.2021.145285. Epub 2021 Jan 22. PMID: 33515893.

Thus, the authors should strengthen their focus on milk miR-mediated regulation of FTO expression affecting pivotal checkpoints of adipogenesis.

Moreover, FTO is also involved in the regulation of beige/brown adipogenesis.

FTO deficiency promotes the browning process of white adipose tissue (WAT). This adaptation appears to be primarily regulated by changes in miR expression.

Ronkainen J, Mondini E, Cinti F, Cinti S, Sebért S, Savolainen MJ, Salonurmi T. Fto-Deficiency Affects the Gene and MicroRNA Expression Involved in Brown Adipogenesis and Browning of White Adipose Tissue in Mice. Int J Mol Sci. 2016 Nov 7;17(11):1851. doi: 10.3390/ijms17111851. PMID: 27827997; PMCID: PMC5133851.

Of importance, FTO expression in peripheral blood mononuclear cells is significantly overexpressed in formula-fed infants compared to breastfed infants.

Cheshmeh S, Nachvak SM, Rezvani N, Saber A. Effects of Breastfeeding and Formula Feeding on the Expression Level of FTO, CPT1A and PPAR-α Genes in Healthy Infants. Diabetes Metab Syndr Obes. 2020 Jun 26;13:2227-2237. doi: 10.2147/DMSO.S252122. PMID: 32617012; PMCID: PMC7326192.

Discussion and Conclusions

The authors should emphasize that milk exosomal miRs affect the magnitude of DNMT1-regulated DNA methylation, FTO-controlled m6A RNA levels and the expression of Wnt proteins and Wnt inhibitors – all critical players determining adipocyte stem cell development and adipogenesis. This important regulatory postnatal system is not provided by artificial formula feeding. Thus, formula is a metabolic violation of the developing infant promoting the epidemic of obesity.

Accumulated evidence unravels breast milk as a human lactation-genome-derived and controlled species-specific programming system of mammalian evolution and not a simple feeding system. Evolutional biology over millions of years guaranteed gene regulatory perfection and adaptations in contrast to human manufactures trying to produce a substitute since 100 years.

The term “breastfeeding” is already a misleading simplification of milk´s true nature. Obviously, pediatricians of the 1920´s made a serious mistake when claiming “milk is just food“.

Bryder L. From breast to bottle: a history of modern infant feeding. Endeavour. 2009 Jun;33(2):54-9. doi: 10.1016/j.endeavour.2009.04.008. Epub 2009 May 21. PMID: 19464060.

Taken together, several important aspects of the programming effects of milk´s miRs affecting brown and white adipogenesis presented in this review require substantial improvements and a deeper consideration of already published information.
